# Core-dependent post-translational modifications guide the biosynthesis of a new class of hypermodified peptides

Zeng-Fei Pei [1], Lingyang Zhu [2] & Satish K. Nair [1,3,4] ✉

The ribosomally synthesized and post-translationally modified peptide (RiPPs) class of natural products has undergone significant expansion due to the rapid growth in genome sequencing data. Using a bioinformatics approach, we identify the dehydrazoles, a novel class of hypermodified RiPPs that contain both side chain dehydration of Ser residues, and backbone heterocyclization at Ser, Thr, and Cys residues to the corresponding azol(in)es. Structure elucidation of the hypermodified peptide carnazolamide, a representative class member, shows that 18 post-translational modifications are installed by just five enzymes. Complete biosynthetic reconstitution demonstrates that dehydration is carried out by an unusual DUF4135 dehydration domain fused to a zinc-independent cyclase domain (CcaM). We demonstrate that CcaM only modifies Ser residues that precede an azole in the core peptide. As heterocyclization removes the carbonyl following the Ser residue, CcaM likely catalyzes dehydration without generating an enolate intermediate. Additionally, CcaM does not require the leader peptide, and this core-dependence effectively sets the order for the biosynthetic reactions. Biophysical studies demonstrate direct binding of azoles to CcaM consistent with this azole moiety-dependent dehydration. Bioinformatic analysis reveals more than 50 related biosynthetic gene clusters that contain additional catalysts that may produce structurally diverse scaffolds.

Ribosomally synthesized and post-translationally modified peptides (RiPPs) are natural products of extraordinary chemical complexity that are derived from linear peptides of ribosomal origin[1,2]. The intricacies of RiPP scaffolds belies their biosynthetic origins as the post-translational modification that elaborate the final mature product often requires very few biosynthetic enzymes[3,4]. For example, the cytotoxic β-helical peptide aeronamide contains 35 post-translational modifications but require only five enzymes to produce the mature final product from a ribosomal peptide precursor[5]. The post-translational modifications imbue the modified peptides with

conformational restraint, resistance to proteases and/or thermal denaturation and often result in final products with biological activities[6].

Two archetypal modifications found in RiPPs are the side chain dehydration of Ser and Thr residues to form the dehydro amino acids dehydroalanine (Dha) or dehydrobutyrine (Dhb) and the backbone heterocyclization of Ser, Thr or Cys residues to form (methyl)oxazolines or thiazolines. Dehydration is catalyzed by one of six known classes of lanthipeptide synthetases, and heterocyclization is carried out by enzymes of the YcaO superfamily[7–9]. Dehydro amino acids are

[1]Department of Biochemistry, University of Illinois at Urbana-Champaign, Urbana, IL 61801, USA. [2]School of Chemical Sciences, NMR Laboratory, University of Illinois at Urbana-Champaign, Urbana, IL 61801, USA. [3]Institute for Genomic Biology, University of Illinois at Urbana-Champaign, Urbana, IL 61801, USA. [4]Center for Biophysics and Quantitative Biology, University of Illinois at Urbana-Champaign, Urbana, IL 61801, USA. ✉e-mail: snair@illinois.edu

widely found in lanthipeptides[10], linaridins[11], thioamitides[12,13], and merchercharmycins[14], either as constituents of the final product or as intermediates for subsequent conjugate addition reactions that produce the final product. Azol(in)es are modifications common in cyanobactins[15], bottromycins[16], linear azol(in)e-containing peptides, among others[17] (Supplementary Fig. 1, 2).

Two main RiPP classes that contain both dehydro amino acids and azol(in)es are the thiopeptides[18] and goadsporin (and the related spongiicolazolicins)[19,20]. In each case, the first modification is the heterocyclization of Ser/Thr/Cys by a YcaO followed by oxidation by a flavin-dependent dehydrogenase to yield the corresponding azole. Subsequently, a class I split LanB-type lanthipeptide dehydratase carries out the dehydration of Ser/Thr using glutamyl tRNA to activate the side chain for subsequent β-elimination[21,22]. As both the YcaO and the LanB function Ser and Thr residues in the peptide substrate, the basis for timing and selectivity between these activities is not yet known. Prior studies on the thiopeptide thiomuricin show that TbtB, a homolog of the LanB glutamylation domain, only modifies the heterocyclized TbtA substrate[21]. However, TbtB only adds a single glutamate out of four possible, and addition of the homolog of the LanB elimination domain (TbtC) is necessary for complete four-fold dehydration of the modified substrate. The crystal structure of TbtB bound to an inert substrate analog does not reveal any obvious active site differences from the glutamylation domain of NisB[23]. Hence, the basis for the preference of TbtB thiopeptide glutamylation enzyme for an azole-modified substrate is unclear.

In this work, we focused on identifying biosynthetic clusters that produce new RiPPs, particularly those encoding for final products that may contain multiple classes of post-translational modifications. A bioinformatic search was directed to identify clusters that contained genes that could catalyze both side chain dehydration and backbone heterocyclization, reasoning that natural products produced by such cluster may contain unexpected scaffolds. Neighborhood network analysis of the ATP-dependent dehydration domains was used to screen for those that were within proximity with genes encoding for YcaO domains. These efforts identified ~50 gene clusters containing a DUF4135, encoding for the dehydration domain of a class II LanM-type lanthipeptide synthetase[24,25], which are syntenic with a gene encoding a YcaO. These presumptive RiPP products (termed dehydrazoles) likely contain both dehydro amino acids and azol(in)es.

Through a series of heterologous co-expression and in vitro characterization, we characterize carnazolamide as a representative member of the dehydrazoles. The corresponding biosynthetic cluster contains a novel DUF4135 with a zinc-independent cyclase domain found in class III LanKCs (hereafter denoted as LanM$_b$C (Supplementary Fig. 3) as these sequences are a fusion of a divergent LanM dehydration domain to a class III cyclase domain), a YcaO/dehydrogenase pair, a radical SAM epimerase and an unusual S-methyltransferase. Together, these five enzymes install a total of 18 modifications, including eight dehydro amino acids, eight azoles, Pro epimerization, and Cys methylation on a 30-residue peptide. Notably, the LanM$_b$C functions independent of the leader sequence and can only modify peptide substrates that have undergone heterocyclization to the azole. Both the DUF4135 domain and the class III cyclase domains of LanM$_b$C are necessary for this activity. Biophysical studies demonstrate the direct binding of the azole moiety by the LanM$_b$C. The modified core-binding sets the biosynthetic timing for reactions, ensuring that the YcaO and LanM$_b$C do not compete for the same substrate. Further bioinformatics analysis reveals additional putative biosynthetic pathways in which the YcaO and LanM$_b$C pair co-occur with other modification enzymes that likely carry out thioamidation or facilitate oxidative crosslinking which expand the chemical space of the dehydrazoles.

## Results

### Genome mining reveals YcaO and DUF4135 hybrid gene clusters
We focused our initial bioinformatic efforts using the sequences of dehydration domains from classes II lanthipeptide synthetases. The EFI-Enzyme Similarity Tool (EST)[26] was used to generate a sequence similarity network (SSN) of DUF4135 (PF13575)[25], which corresponds to the dehydration domain of the class II LanM. Using the UNIPROT database, a total of 3277 sequences (as of Jan 2023) were utilized to create an SSN with an alignment score of 80. Sequences were confined to a minimum length of 500 residues to minimize potential false positives (typical of the LanM dehydration domain) (Fig. 1a). The resultant SSN was ported to the EFI-Genome Neighborhood Tool[27] to query the ten open reading frames that flank each sequence. The resultant genome neighborhood diagrams (GNDs) showed that six networks consist of a class II LanM sequence that co-occurs with a sequence for a YcaO (PF02624) (Fig. 1b). A total of 49 such nodes were identified in this analysis, and correspond to networks 5, 36, 39, 42, 55 and 59 in Fig. 1b. Identification of a previously reported BGC containing both a YcaO and a LanM in network 36 demonstrates the feasibility of our strategy[24]. The primary sequences of all LanM in these clusters consist of a DUF4135 domain fused to a class III cyclase domain that lacks the conserved residues that are involved in binding the requisite zinc ligand (Supplementary Fig. 4). Hence, these LanM$_b$C sequences represent a novel class of dehydratases.

We next analyzed network 5 as it contains the most abundant number of gene clusters with syntenic LanM$_b$C and YcaO (38 in total). All other networks contain less than 5 gene clusters each. A candidate cluster (cca; Fig. 1c) from network 5 is from the aerobic, mesophilic Flavobacteriia Chryseobacterium carnipullorum DSM25581. The biosynthetic cluster encodes a YcaO (CcaB, WP_073335636.1), a LanM$_b$C (CcaM, WP_073335633.1, Fig. 1d and Supplementary Fig. 5), along with a putative precursor peptide (CcaA, WP_073335637.1) that has a Cys rich C-terminus and a large N-terminal nitrile hydratase-like leader peptide (NHLP)[28]. Hence, the product of the biosynthetic cluster is likely a new subfamily of NHLP-derived proteusins[29,30]. Additional genes in the cluster include a dehydrogenase that likely oxidizes the YcaO-derived azolines (CcaC, WP_167367437.1), a radical SAM epimerase (CcaD, WP_073335634.1), a SAM-dependent methyltransferase (CcaH, WP_073335629.1) and a S8 peptidase (CcaF, WP_167367436.1).

### Reconstitution studies of cca biosynthetic gene cluster
We first attempted to isolate the final mature hypermodified peptide from cultures of C. carnipullorum DSM25581. However, despite exhaustive attempts using different growth conditions and mediums, no products could be observed by liquid chromatography-mass spectrometry (LC-MS) within the expected mass ranges. We next pursued a heterologous co-expression strategy using Escherichia coli BL21(DE3) that proved to be fruitful. Specifically, the N-terminally polyHis tagged precursor peptide (CcaA) was co-expressed with a combination of untagged biosynthetic enzymes and purified under denaturing conditions, treated with tobacco etch virus (TEV) protease to remove the affinity tag, and further purified using high-performance liquid chromatography (HPLC). Matrix-assisted laser desorption/ionization coupled to time-of-flight mass spectrometry (MALDI-TOF MS) analysis was used to monitor the extent of modifications installed.

Initial efforts were directed at elucidating the function of the YcaO protein (CcaB) and the associated dehydrogenase (CcaC). Full-length unmodified precursor (CcaA) could not be purified in sufficient quantities for complete in vitro analyses. Hence, we focused efforts on co-expression studies. Co-expression of CcaA with the CcaB and CcaC resulted in a mass loss of 160 Da, consistent with the formation of eight azoles (peptide CcaA$^{BC}$, Fig. 2). These data showed that out of a total of 20 Ser, Thr, or Cys residues in the core peptide, only eight were subject to heterocyclization. Additional co-expression of the potential epimerase CcaD together with CcaB/CcaC pair resulted with no further

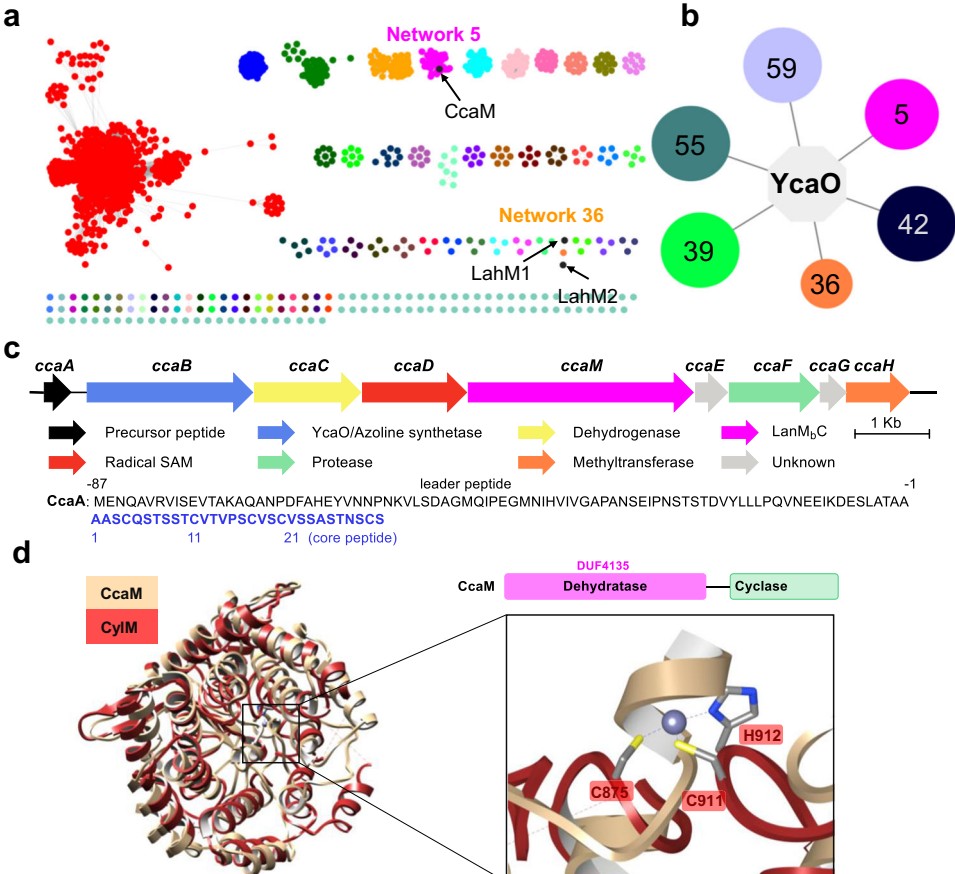

**Fig. 1 | Bioinformatic analysis of LanM and YcaO hybrid biosynthetic clusters.** **a** A sequence similarity network (SSN) of DUF4135 (PF13575). **b** Genome neighborhood network (GNN) of Pfam family hub nodes (YcaO, PF02624) with SSN cluster-spoken nodes colored to correspond to the coloring in panel **a**. **c** A LanM$_b$C and YcaO hybrid biosynthetic gene cluster from *C. carnipullorum* DSM25581. The sequence of the precursor peptide is shown, with the leader peptide colored in black, and the core peptide colored in blue. **d** Structural superposition of the cyclase domain of CylM and predicted CcaM. The zinc-binding site is not conserved in CcaM.

mass change as expected for the potential epimerization reaction (Fig. 2). Co-expression of the SAM-dependent methyltransferase CcaH together with CcaB/CcaC or CcaB/CcaC/CcaD resulted in another +14 Da mass increase suggested one methylation on the peptide with eight azoles (Fig. 2). Interestingly, co-expression of CcaA with CcaH resulted in only +14 Da mass increase, suggesting a single (regioselective) methylation on the unmodified precursor peptide (Fig. 2).

We next focused on the novel LanM$_b$C (CcaM) which is predicted to carry out dehydration of Ser and Thr residues. Co-expression of the CcaA precursor peptide with CcaM alone did not result in any modifications. This was unexpected as the class II DUF4135 domain uses ATP as a cofactor[25] and should be functional in heterologous *E. coli* expression. However, co-expression of CcaA with CcaM, along with CcaB/CcaC/CcaD, resulted in a series of additional −18 Da mass shifts (up to −144 Da) in azole containing precursor peptide (peptide CcaA$^{BCDM}$, Fig. 2), consistent with up to eight dehydrations. These observations suggest that the CcaM dehydratase only functioned on the peptide that had previously undergone heterocyclization. In addition, co-expression of CcaA with CcaB/CcaC/CcaD, CcaM, as well as with CcaH methyltransferase resulted in another +14 Da mass increase consistent with the methylation (Fig. 2).

Next, we attempted to elucidate the function of the presumptive radical SAM (rSAM) epimerase CcaD. In canonical proteusins, homologous rSAM (such as PoyD from the polytheonamide pathway; 28% sequence identity) (Supplementary Fig. 6) catalyze abstraction of the Cα hydrogen, followed by hydrogen atom transfer to the radical intermediate to produce an epimerize residue in the precursor peptide (and a protein thiyl radical)[31,32]. Co-expression of the CcaA precursor with CcaD yielded only minute amounts of product (peptide CcaA$^D$), and we were unable to purify enough peptide for detail characterization using the orthogonal D$_2$O-based induction system (ODIS) developed by the Piel lab[33]. Marfey's analysis of various intermediates and the final product demonstrate that CcaD carries out epimerization only after all other modifications have been installed (see next section).

To identify the position of each modification and characterize the final product, the leader peptide must be excised. To this end, we focused on the putative peptidase CcaF. The sequence of CcaF is annotated as a S8 (subtilisin type) serine protease and contains a classical Asp-His-Ser catalytic triad[34]. Recombinant CcaF was heterologously expressed in *E. coli* BL21(DE3) and purified as a polyHis-tagged protein (Supplementary Fig. 7). Incubation of the CcaA$^{BC}$ peptide (containing eight azoles) with CcaF resulted in facile removal of the leader peptide (89 residues including the N-terminal Ser-Gln after TEV digestion) and revealed that the cleavage site occurs between four consecutive Ala (Supplementary Fig. 8). Similarly, incubation of the full-length peptide products co-expressed with different modification enzymes including CcaA$^{BCH}$, CcaA$^{BCDH}$, CcaA$^{BCDM}$ and CcaA$^{BCDMH}$ with CcaF resulted in removal of the leader peptide (Supplementary Fig. 8). Last, a 10-mer peptide encoding the last 8 residues of the leader peptide and the first two alanines of core peptide was synthesized and incubated with CcaF, the short peptide was also digested at the expected position suggesting a short recognition sequence of the protease (Supplementary Fig. 9). Given the structural divergence

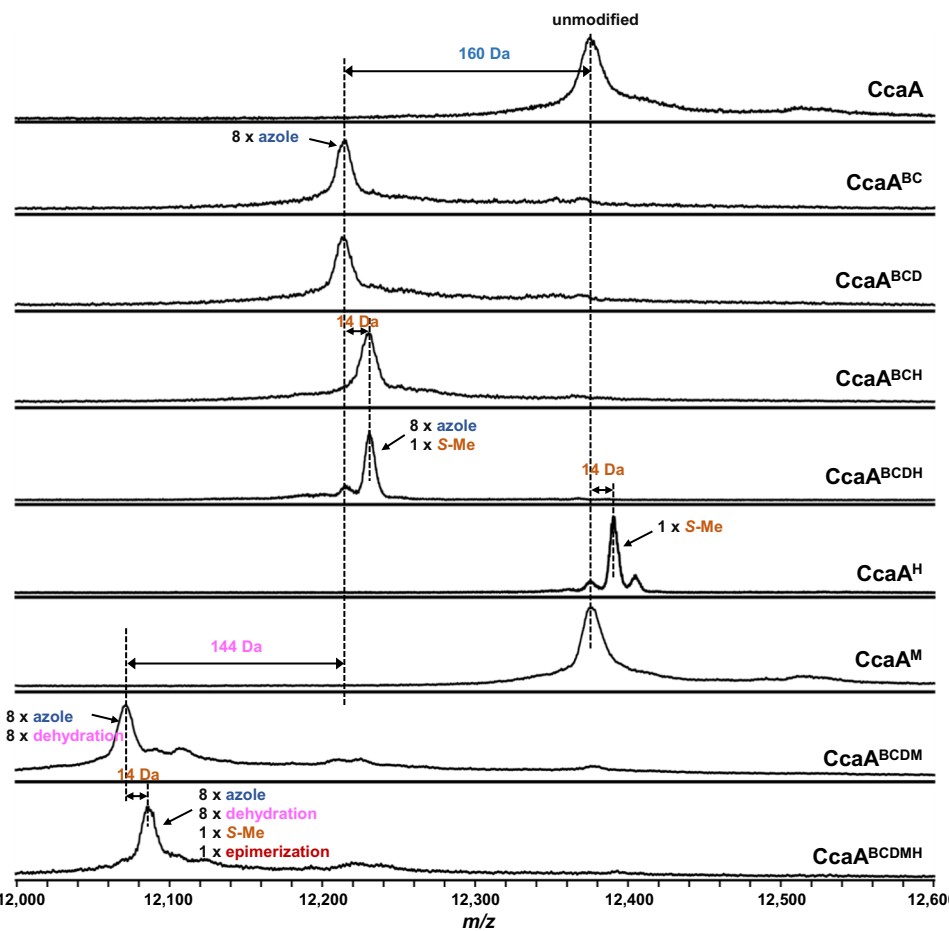

**Fig. 2 | MALDI-TOF MS analysis of heterologous co-expression.** MALDI-TOF mass spectra of purified full-length precursor peptide that were co-expressed with different processing enzymes from *cca* biosynthetic gene cluster in *E. coli*.

amongst these modified peptides, CcaF appears to be a protease with a broad substrate scope.

## Structural elucidation of carnazolamide

To facilitate structural elucidation, we first scrutinized the azole-containing intermediate produced by co-expression of CcaA with CcaB/CcaC/CcaH, followed by in vitro removal of the leader peptide by CcaF. The modified core peptide (cCcaA$^{BCH}$) was purified by HPLC and analyzed by MALDI-TOF MS and MS/MS, iodoacetamide (IAA) labeling and multi-dimensional NMR spectroscopy (Fig. 3a, Supplementary Figs. 10, 11 and Table 4). These data show that the structure of cCcaA$^{BCH}$ consists of eight azoles, including four thiazoles, one oxazole and three methyloxazoles together with an unusual methylated Cys (Fig. 3c). These analyses also demonstrate that CcaB is not chemoselective, as Cys, Thr, and Ser residues in the core peptide are each modified to the azoline.

We next focused on characterization of the presumptive final product (modified core peptide cCcaA$^{BCDMH}$, termed carnazolamide) through co-expression of the CcaA precursor with the entire set of modification enzymes (CcaB/CcaC/CcaD/CcaM/CcaH), followed by in vitro removal of the leader peptide with CcaF, HPLC purification and analysis using MALDI-TOF MS and MS/MS, and multi-dimensional NMR spectroscopy (Fig. 3a, Supplementary Figs. 12, 13 and Table 5). These data show that carnazolamide (Fig. 3c) contains an additional eight Dha as compared with the structure of cCcaA$^{BCH}$. Of note, all the Dha residues are located right before an azole while Ser8, Thr13, and Ser30 were left intact. Indeed, CcaA can be mono-methylated by CcaH with or without other modifications

present suggesting that *S*-methylation is independent of other modifications.

To establish the identity and nature of the residues modified by the CcaD epimerase, we carried out Marfey's analysis on the eight azolic core peptide (cCcaA$^{BCH}$), the core peptide when co-expressed with CcaB/C/H and the epimerase CcaD (cCcaA$^{BCDH}$), and the final product carnazolamide. First, comparison of cCcaA$^{BCH}$ and cCcaA$^{BCDH}$ shows that they are identical, and all the residues are L-configuration (Supplementary Fig. 14). This suggests that the epimerase does not function on unmodified peptide, or the peptide modified with only the azoles. Analysis of the final product carnazolamide shows that Pro15 is largely in the D-configuration while other residues are in the L-configuration (Fig. 3b and Supplementary Fig. 15), suggesting that the CcaD rSAM epimerase functions late in the biosynthetic process after the azoles and dehydro amino acids have been installed. The timing and nature of the epimerization catalyzed by CcaD contrasts with those catalyzed by homologs found in the polytheonamides and aeronamide A. In these other proteusins, epimerization can be an early-stage modification as the epimerase can modify multiple sites on their cognate unmodified precursor peptides[3,35]. Hence, carnazolamide is a hypermodified peptide with 18 modifications that represents a new sub-clade of the proteusins.

We next tested purified carnazolamide for biological activity using standard assays and methodologies. However, carnazolamide did not demonstrate any antibacterial activities and cytotoxicity activities against tested strain (Supplementary Table 6). Continued efforts are ongoing to test the ecological and/or physiological function of this and other related modified peptides.

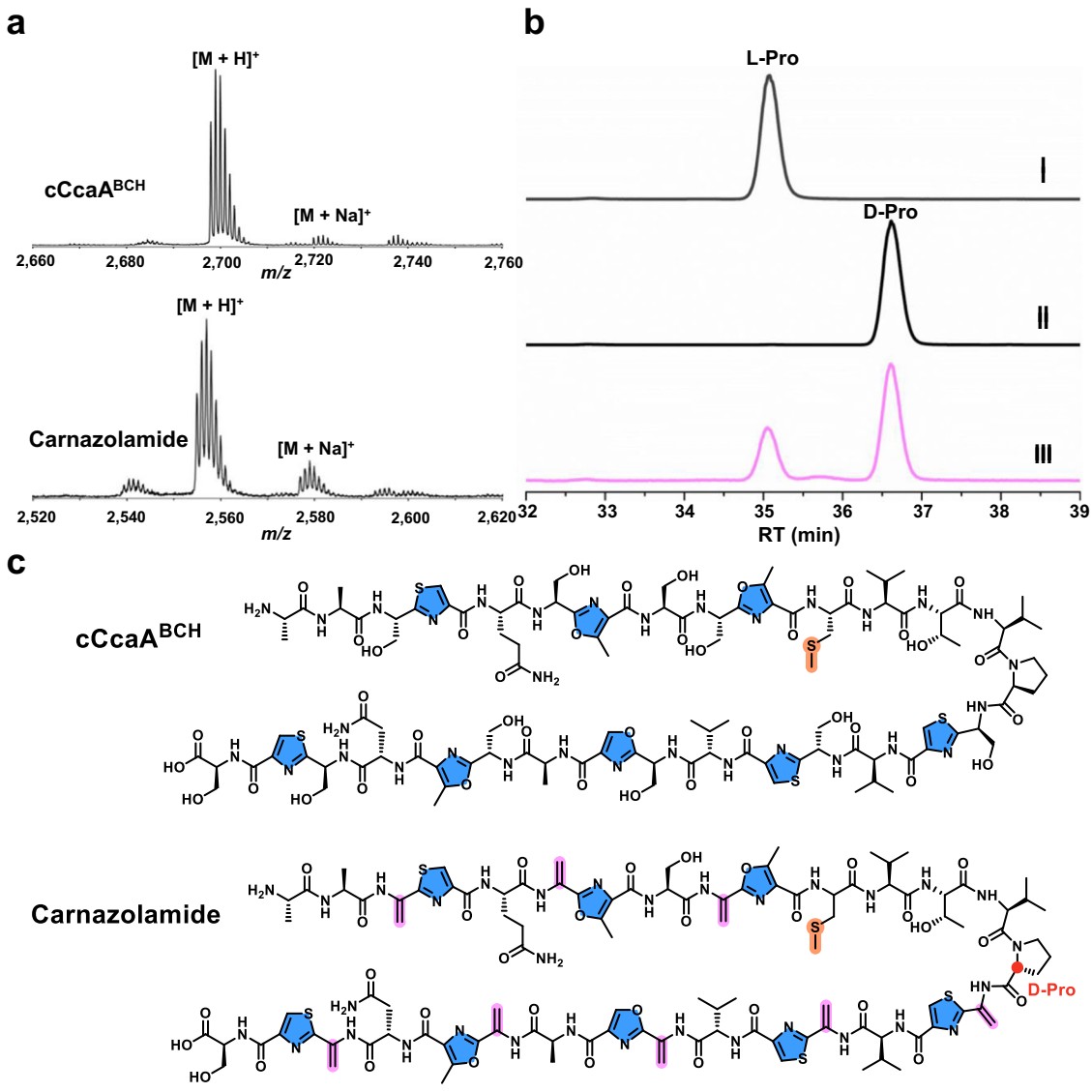

**Fig. 3 | Structural elucidation of cCcaA^BCH and carnazolamide. a** MALDI-TOF mass spectra of cCcaA^BCH (exact mass calculated $m/z$ = 2,698.0; observed 2,698.2) and carnazolamide (exact mass calculated $m/z$ = 2,553.9; observed 2,554.1). **b** Marfey's assay of carnazolamide hydrolysate, extracted ion chromatogram ($m/$ $z$ = 368) of FDAA derivatized proline was shown. L-proline standard (I), D-proline standard (II) and carnazolamide hydrolysate (III). **c** Structure of the azole containing intermediate cCcaA^BCH and the presumptive final product carnazolamide. FDAA, 1-fluoro-2-4-dinitrophenyl-5-L-alanine amide.

## Biochemical characterization of the core-dependent dehydratase CcaM

Our co-expression studies suggested that CcaM does not function on the unmodified CcaA precursor peptide only functions on the substrate that has been modified with the installation of azoles. We sought to confirm this by reconstituting the CcaM-catalyzed dehydration reaction in vitro. CcaM was heterologously expressed in *E. coli* BL21(DE3) and suitable quantities of unmodified CcaA was also produced in *E. coli* through large-scale fermentation (Supplementary Fig. 7). As expected, incubation of recombinant CcaM with unmodified CcaA precursor peptide in the presence of ATP and MgCl₂ did not result in any mass shifts as determined by MALDI-TOF MS (Supplementary Fig. 16). However, incubation of the full-length CcaA^BCH peptide (containing eight azoles and Cys methylation) resulted in up to eight dehydrations, with the seven-dehydrated product as the major species (Fig. 4a). These data confirm that CcaM functions only on the azole modified peptide and defines this enzyme as a novel core-dependent dehydratase.

The observation that CcaM (and likely other LanM_bC) only functions on modified core peptides raised the possibility that such enzymes may be independent of the leader peptide. To test this hypothesis, we incubated the core peptide cCcaA^BCH with CcaM in the presence of ATP and MgCl₂, and observed up to eight dehydrations, as determined by MALDI-TOF MS (Fig. 4b). These data suggest that CcaM catalyzed dehydration is independent of leader peptide. The tri-, hexa- and hepta-dehydrated intermediates were isolated by HPLC, and MALDI-TOF MS/MS analysis of these intermediates revealed that dehydration occurs with roughly N to C directionality (Supplementary Fig. 17).

As noted, the carnazolamide product that was isolated by co-expression with the entire biosynthetic pathway contained eight dehydro amino acids but lacked any lanthionine rings. As the class III cyclase domain is seemingly unnecessary, we sought to determine if the DUF4135 domain alone could support dehydration. Prior studies demonstrate that the excised dehydration domains of CylM[36], HalM2[37], and NukM[38] (all class II), of CurKC (class III)[39] and of SgbL (class IV)[40,41] were sufficient for dehydration. We individually expressed and purified the N-term dehydration domain CcaM_N (residues 1-645) and C-term class III cyclase domain CcaM_C (residues 645-1015) (Supplementary Fig. 7). Incubation of the azole-containing peptide cCcaA^BCH with either

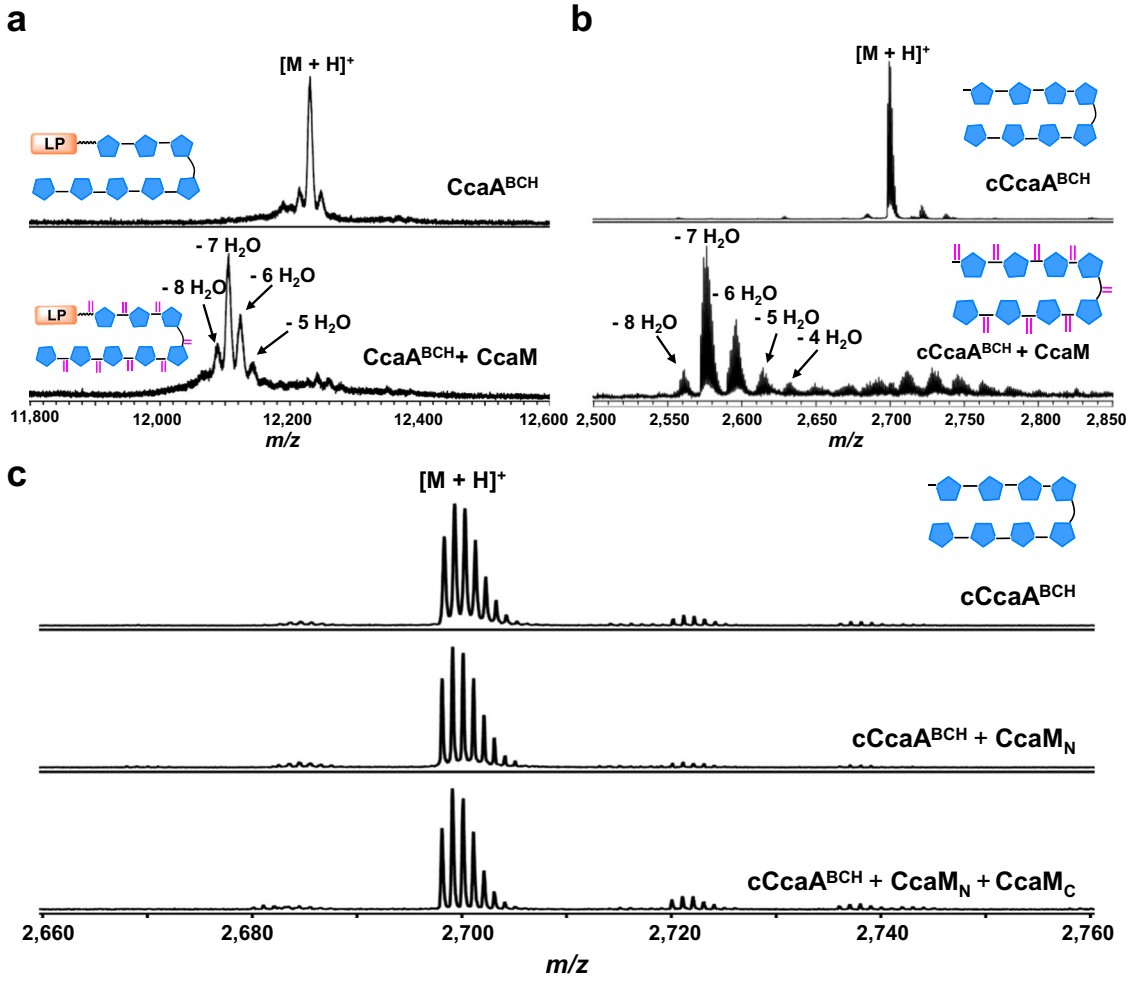

**Fig. 4 | In vitro reconstitution of CcaM activity. a** Enzymatic assay of CcaM with octa-azole containing full length precursor peptide CcaA^BCH. **b** Enzymatic assay of CcaM with octa-azole containing core peptide cCcaA^BCH. **c** Enzymatic assay of CcaM_N and CcaM_N/CcaM_C with octa-azole containing core peptide cCcaA^BCH. LP leader peptide.

the CcaM_N dehydration domain or the combination of CcaM_N and CcaM_C in the presence of ATP and MgCl_2 resulted in no modifications as determined by MALDI-TOF MS (Fig. 4c). Hence, the class III cyclase domain is necessary for the dehydration activity of CcaM and requires a fused dehydration domain, as the individual domains do not function in trans.

## Azole moiety-dependent dehydration of CcaM

Having shown that the CcaM is leader-independent and only acts on the azole-modified core peptide, we next investigated the substrate tolerance of this enzyme. We generated four precursor peptides consisting of core residues Ala1 to Thr10, to Cys20, to Ser23, and to Thr26 and co-expressed each of these with CcaB/CcaC/CcaH. Each peptide was affinity purified, treated with CcaF to remove the leader, and further purified by HPLC. Of note, the CcaB YcaO domain did not modify any C-terminal Cys/Ser/Thr, as determined by MALDI-TOF MS/MS, suggesting that it may require a follower sequence (Supplementary Fig. 18).

Each of the four azole-containing truncated core peptides was incubated with CcaM, ATP and MgCl_2 and in vitro dehydration was monitored by MALDI-TOF MS. Unexpectedly, all these substrates could be processed by CcaM, including the shortest 10-residue peptide (Figs. 5a–d). Of note, all the peptides had one fewer dehydration than expected, probably due to the lack of the azole at the C-terminal residue. To test this hypothesis further, we carried out large-scale

purifications of each of the peptides and characterized the CcaM modified products by tandem mass spectrometry (Supplementary Fig. 18). These data demonstrated that the missing dehydration each occur at the residue preceding the missing azole, consistent with the observation that CcaM only processes Ser residues preceding an azole.

To rule out regioselectivity as a factor for the lack of the terminal dehydration on each of the short peptides, we constructed the Thr7→Ala variant of CcaA. This variant peptide would be expected to lack one azole internal in the peptide at position 7 when co-expressed with CcaB/CcaC/CcaH. The hepta-azolic core peptide cCcaA(T7A)^BCH was HPLC purified, and its identity confirmed by MALDI-TOF MS/MS (Supplementary Fig. 19). Incubation of this peptide with CcaM, ATP and MgCl_2 resulted in up to five dehydrations, and tandem mass spectroscopic analysis of the penta-dehydrated product confirmed that Ser6 was not modified (Fig. 5e). These data confirm that CcaM only carries out dehydrations at Ser residues preceding an azole, regardless of its location in the precursor peptide.

The observation that CcaM only modified Ser residues that precede an azole suggests that the LanM_bC enzymes are mechanistically distinct from canonical class I LanB, class II LanM, class III LanKC, and class IV LanLs. In each of the latter classes of enzymes, prior studies are consistent with a mechanism in which removal of the α-proton from the phosphorylated Ser or Thr residue yields an enolate intermediate that can undergo β-elimination to form the Dha or Dhb, respectively. In the case of CcaM (and presumably other LanM_bCs), the residue

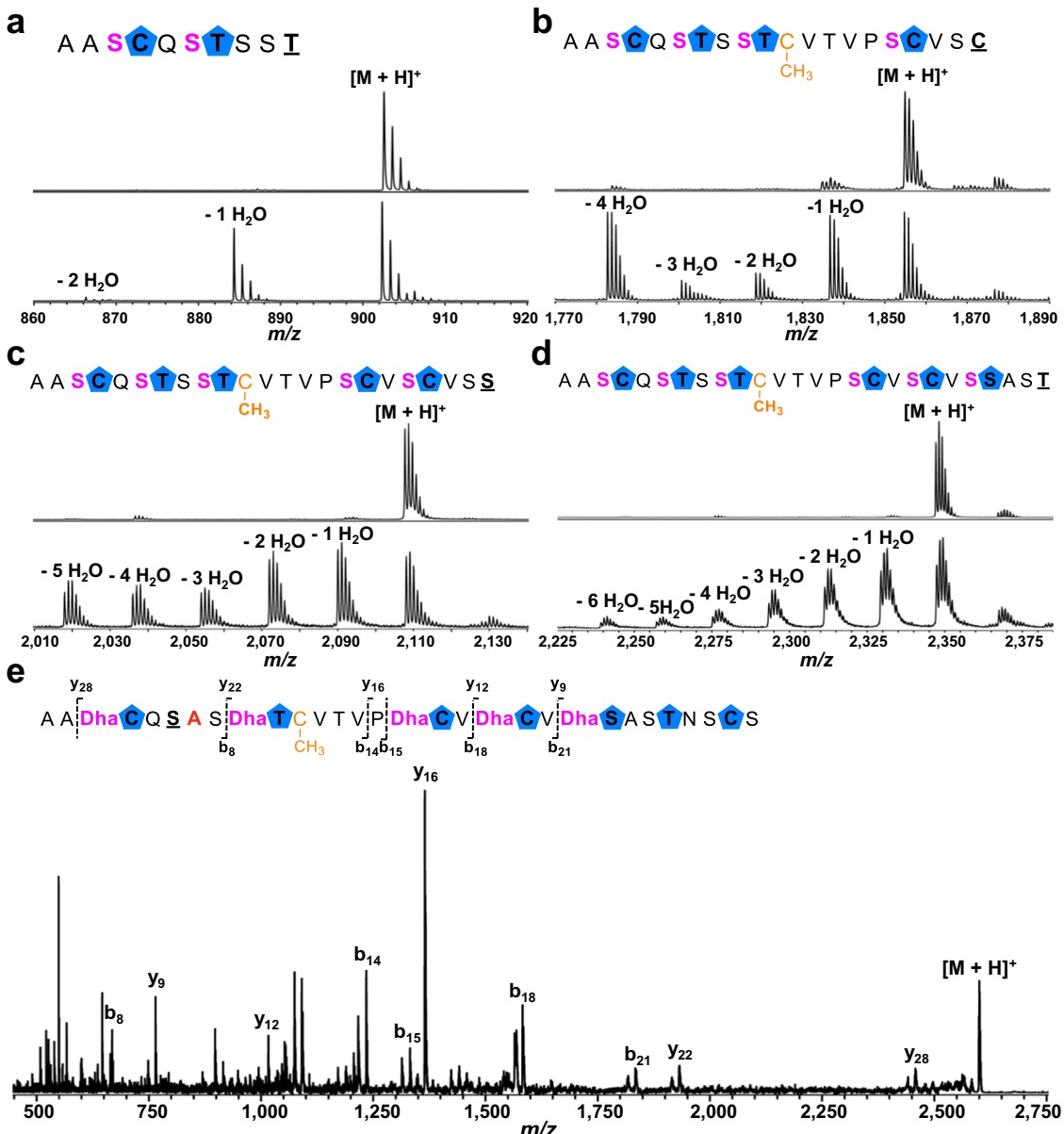

**Fig. 5 | Substrate tolerance of CcaM.** In vitro CcaM assay with different truncated azole containing core peptides, no enzyme control (top), CcaM assay (bottom). Different substrates were shown in cartoon, cCcaA(1-10)$^{BCH}$ **a**, cCcaA(1-20)$^{BCH}$ **b**, cCcaA(1-23)$^{BCH}$ **c**, and cCcaA(1-26)$^{BCH}$ **d**, the last underlined amino acid means it is not heterocyclized by YcaO, serine that undergoes dehydration by CcaM is shown in pink. **e** MALDI-TOF MS/MS of in vitro penta-dehydrated product of CcaM assay with hepta-azole containing core peptide cCcaA(T7A)$^{BCH}$.

following the Ser is heterocyclized and lacks a carbonyl, precluding enolate formation. Thus, it is likely that for these enzymes, the phosphorylated Ser undergoes β-elimination without the formation of an enolate intermediate (Supplementary Fig. 20).

**Biophysical investigation of azole moiety binding to CcaM**

To further characterize the interaction between azole-containing core peptides, we sought to use fluorescence polarization (FP) assay to monitor substrate binding to CcaM. We labeled both the modified full-length precursor peptide variant CcaA(M-87C)$^{BCH}$ with 5-IAF (5-Iodoacetamidofluorescein) on Cys-87, as well as the N-terminus of the modified core peptide cCcaA$^{BCH}$ with FITC (fluorescein isothiocyanate). Despite numerous attempts under varying conditions, none of the fluorescence polarization (FP) binding curves reached saturation, ruling out the use of FP to measure binding. We were able to overcome experimental challenges by measuring the binding of

cCcaA$^{BCH}$ to CcaM using saturation transfer difference nuclear magnetic resonance (STD-NMR) spectroscopy. The STD-NMR experiment is based on the nuclear Overhauser effect (NOE) and directly measures magnetization transfer from the enzyme to the ligand[42]. Observation of the ligand resonance signals can be used to identify hydrogens in the peptide ligand that are close to the protein upon binding.

The STD-NMR experiment was carried out on a sample containing CcaM and a core peptide containing the eight azoles cCcaA$^{BCH}$ by selectively irradiating at $^1$H −1.0 ppm with a 3.0 s saturation time at 25 °C. This spectrum was subtracted from the reference spectrum (1D proton spectrum of cCcaA$^{BCH}$ irradiating at 30 ppm) giving the STD NMR difference spectrum. Clear observation of signals corresponding to residues 11 through 29 could be observed demonstrating that the cCcaA$^{BCH}$ core peptide binds to CcaM (Fig. 6 and Supplementary Table 4). Interestingly, the C-terminus of the peptide binds slightly tighter than the N-terminus as more signals were observed for the

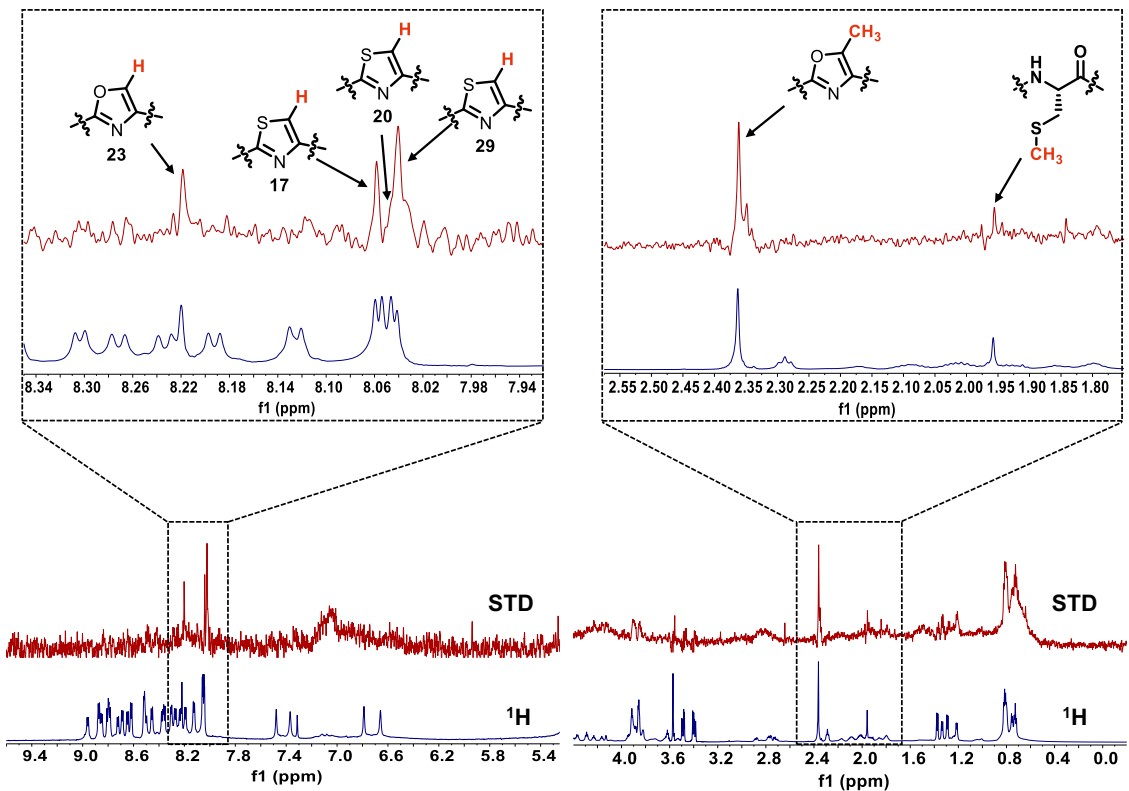

**Fig. 6 | STD-NMR of cCcaA^BCH with CcaM.** Corresponding STD spectrum (top) and reference ^1H spectrum of cCcaA^BCH (bottom). Representative protons interacting with CcaM were shown, the protons on the azole moieties demonstrated direct interaction between azoles and CcaM.

C-terminus. The hydrogens on thiazoles (derived from Cys17, Cys20 and Cys29) and oxazole (Ser23) and methyl group of methyloxazole were observed. Due to signal overlap of the methyl group of methyloxazoles, the signal at 2.36 ppm signal could not be confidently assigned as rising from only one methyloxazole or all the three methyloxazoles. Nevertheless, these STD signals clearly show that CcaM directly binds to the azole moieties, consistent with the in vitro assay results showing that dehydration only occurs at Ser residues that precede an azole.

**Bioinformatic expansion reveals novel dehydrazole pathways**
With the data from biosynthetic reconstitution studies of the *cca* cluster on hand, we further probed the biosynthetic landscape of other gene clusters containing both a LanM_bC and a YcaO. Our analysis shows that homologous clusters are widely distributed across multiple phyla but are found mostly in Proteobacteria and Actinobacteria. Each of these clusters contains genes that may encode precursor peptide consisting of a long NH-like leader peptide and core sequences that are enriched with Ser, Thr, and Cys residues. A few representative clusters that likely produce novel final products are detailed (Supplementary Fig. 21).

The cluster from uncultured *Candidatus thioglobus* sp. contains genes for two YcaO/dehydrogenase pairs but only one LanM_bC. The cluster also encodes for multiple precursor peptides with highly divergent leader peptides. Conceivably, each of the two YcaOs work on different precursor peptide, while the LanM_bC, which is leader independent, can carry out dehydration of both peptides. A second putative cluster of some interest is found in *Streptomyces* sp. SN-593 and contains the expected LanM_bC and the YcaO and dehydrogenase, as well as a cytochrome P450-like (CYP) sequence and two genes of unknown function. Of note, recent studies have identified RiPP-associated CYPs that are involved in side chain crosslinking[43] or oxidative decarboxylation[44]. A cluster from *Nakamurella* sp. PAMC28650

harbors a second YcaO adjacent to a TfuA, which are genes associated with backbone thioamide formation[45,46]. The precursor peptide in this cluster contains a putative core sequence with multiple adjacent Ala residues, reminiscent of a similar feature in thioviridamide that is the site for thioamide installation[47,48]. Lastly, a cluster found in *Cystobacter fuscus* DSM2262 contains a gene encoding a homolog of Ser/Thr kinases, as found in class III and IV lanthipeptide synthetases[49,50].

## Discussion
The discovery of dehydrazoles described in this work highlights a novel class of RiPP natural products that are elaborated by a newly discovered and widely distributed biosynthetic pathway. A central component of these pathways is the azole-dependent lanthipeptide dehydratase that is functionally and, presumably, mechanistically distinct from other known lanthipeptide synthetases. Characterization of carnazolamide, a representative product from this RiPP class, shows that the 30-residue peptide contains 18 post-translational modifications. Remarkably, all these modifications are installed by just five biosynthetic enzymes (CcaB, CcaC, CcaD, CcaM and CcaH), and a leader protease (CcaF) generates the modified product. Detailed analysis of carnazolamide shows that it contains oxazoles, methyloxazoles, and thiazoles, derived from these Ser, Thr, and Cys, respectively, as well as dehydroalanines derived from Ser. Based on the nature of the long NH-like leader peptide, we suggest that the dehydrazoles may be classified as a sub-clade of the proteusins.

One feature of the dehydroazoles is the presence of both multiple dehydro amino acid and multiple azoles. This adds to the growing list of RiPPs with both modifications, which includes thiopeptides[18] and goadsporin[22]. However, dehydrazoles are distinguished by their biosynthetic origin and are assembled via a novel enzymatic route described here. The most intriguing of these is the newly discovered class of the LanM_bC hybrids consisting of a LanM (class II) dehydratase domain fused to a zinc-independent (class III) cyclase domain. Atypical

of RiPP biosynthetic enzymes in general, the LanM$_b$C does not require the leader peptide, is core-dependent, and can carry out dehydrations on short peptides that contain azole moieties.

Analysis of the substrate scope of CcaM demonstrates that the enzyme only catalyzes dehydration at Ser residues that precede an azole. Characterization of the substrate scope of the enzyme demonstrates that CcaM only catalyzes dehydrations at Ser residues that precede an azole and biophysical characterization using STD-NMR confirms a direct interaction. As the azole following the Ser substrate lacks the carbonyl, the LanM$_b$C likely cannot proceed via the intermediacy of an enolate and is mechanistically distinct from canonical classes I-V lanthipeptide synthetases. The core dependence is important as it sets the biosynthetic timing of the two modifications, as otherwise both the YcaO and the LanM$_b$C would compete for the same substrate. As the latter enzyme only functions on the substrate modified by the former enzyme, this competition is derailed, and the order of the biosynthetic reactions is set.

In principle, the biosynthetic enzymes involved in the formation of thiopeptides or goadsporins must also avert substrate competition. In the case of the thiopeptide thiomuracin, the glutamylation enzyme TbtB only functions on the azole-modified peptide substrate[21]. However, the reaction stalls at one glutamylation (at Ser14) and requires the addition of the elimination enzyme TbtC to complete the four-fold dehydration process. Both the leader peptide and the azole-modified core peptide each bind to TbtB with modest affinity (K$_d$ of ~17 µM and ~0.8 µM respectively). However, the full-length azole-modified peptide binds much more tightly (K$_d$ of ~27 nM)[51]. Lastly, the crystal structure of TbtB shows a canonical NisB-like active site and does not show any features that would suggest core recognition[23]. These data are consistent with the proposed scheme of TbtB recognizing a conformation of the core peptide that is induced by installation of the six azoles[51]. In contrast, CcaM shows direct binding to the azole moiety and can modify even short azolic peptides in the absence of a leader sequence.

The pathway described in this work is widely distributed and there are at least 50 other putative biosynthetic gene clusters in which genes for a YcaO and a LanM$_b$C are co-localized. The precursor peptides for each of these pathways are enriched in Ser, Thr, and Cys residues in the presumptive core sequences. Several of these other clusters also contain genes that are homologous to known RiPP enzymes, including those that catalyze other backbone, and side-chain modifications, while other clusters contain genes with no known homologs. Some of these biosynthetic gene clusters are found in characterized but as-yet uncultured organisms. Our in vivo and in vitro methods described here provide a rational platform for accessing this rich, untapped reservoir of novel natural products.

## Methods
### General materials and methods
Primers and enzymes used for molecular biology were purchased from Integrated DNA Technologies Inc. (Coralville, IA) and New England Biolabs (Ipswich, MA), respectively. Reagents, buffer components and LB were purchased from Fischer Scientific. Agarose gel extraction and plasmid isolation were conducted using spin columns according to the manufacturer's protocol (Syd Labs, MA). DNA sequencing was performed by ACGT Inc. (Wheeling, IL). Isopropyl β-D-1-thiogalactopyranoside (IPTG) and antibiotics were purchased from Gold Biotechnology. E. coli DH10B chemically competent cell was used for cloning and plasmid propagation, while E. coli BL21(DE3) was used for peptide expression, co-expression, and protein expression.

### Plasmid construction
The TEV site was engineered into standard pRSFDuet-1 (kanamycin resistance), pCDFDuet-1 (spectinomycin resistance) and pETDuet-1 (ampicillin resistance) vectors using primers (Duet-TEV-FP and Duet-TEV-RP) ordered in this study (Supplementary Table 3). The genes

(ccaA, ccaB, ccaC, ccaD, ccaM, ccaF and ccaH) were amplified from genomic DNA (gDNA) extracted from Chryseobacterium carnipullorum DSM25581. PCR products and restriction enzyme linearized vectors were separated by agarose gel electrophoresis and purified by spin columns. The PCR product and vector were assembled using the NEBuilder HiFi assembly kit following the manufacturer's guidance. The assembled mixture was transformed into chemically competent cell E. coli DH10B and selected on LB agar plates supplemented with antibiotic respectively. Single colonies were inoculated in LB supplemented with the appropriate antibiotic and grown at 37 °C overnight. The plasmids were isolated, and sequencing was performed to verify the presence of the DNA insert.

### Site-directed mutagenesis of CcaA
CcaA variants were obtained by site-directed mutagenesis of pCcaAD-RSF or pCcaAH-RSF as templates using the QuickChange method with the primers listed in Supplementary Table 3. PCR reactions were performed using NEB Phusion High-Fidelity DNA polymerase or NEB Q5 High-Fidelity DNA polymerase according to the recommended protocol. The desired PCR products were separated by agarose gel electrophoresis and purified by spin column. The purified PCR products were transformed into chemically competent cell E. coli DH10B and selected on LB agar plates containing 50 µg/mL kanamycin. Single colonies were grown in LB with kanamycin at 37 °C for 12 – 16 h; the plasmids were purified, and the inserts were verified by sequencing.

### Heterogeneous expression and purification of full-length modified CcaA
Different combinations (listed in Supplementary Table 2) of the constructed plasmids were co-transformed into E. coli BL21(DE3). Colonies were selected on LB agar plates containing the appropriate antibiotics listed for each plasmid. A single colony was selected and inoculated into 50 mL of LB with the appropriate antibiotics and grown at 37 °C overnight. The overnight cultures were used to inoculate 1.5 L of LB medium with the appropriate antibiotics which were incubated at 37 °C with shaking (150 rpm) until the OD$_{600}$ reached ~ 0.6. Cell cultures were cooled on ice for 30 min before adding IPTG to a final concentration of 0.4 mM, then, the cultures were placed at 18 °C and shaken at 150 rpm for 16 - 20 h. Cells were harvested by centrifugation at 4000× g for 15 min. The supernatant was decanted, and the cell pellets were used for the following peptide purification. Note: when the radical SAM protein CcaD was used in the co-expression system, pACYC-sufABCDSE[52] (Chloramphenicol resistance) was also used and 0.5 mM (NH$_4$)$_2$Fe(SO$_4$)$_2$ and 0.5 mM cysteine were added right after the addition of IPTG.

Cell pellets from 1.5 L cultures were suspended in 30 mL of lysis buffer (50 mM Tris-HCl, 150 mM NaCl, 8 M urea, pH 8.0) and lysed by sonication. Denaturing buffer was used to eliminate any processing enzymes that may be bound to the processed precursor. The cell lysate was centrifuged at 14,000× g for 1 hour at 4 °C, and then the supernatant was applied to 4 mL Ni-NTA agarose resin and washed with 8 mL lysis buffer prior to the elution with 5 mL of elution buffer (50 mM Tris-HCl, 150 mM NaCl, 300 mM imidazole, 8 M urea, pH 8.0). Peptides eluted from the Ni-NTA resin were dialyzed using Spectra/Por™ Dialysis Tubing 3500 Dalton MWCO into dialysis buffer (25 mM Tris-HCl, 50 mM NaCl, pH = 8.0) at 4 °C overnight. The dialyzed peptides were concentrated using 10 kDa MWCO centrifugal filters. The presence of the modified peptides was monitored by MALDI-TOF MS.

For TEV-cleaved peptides, TEV digestion was performed in 50 mM Tris-HCl buffer (pH = 8.0) at RT for 12 h, followed by another Ni-NTA purification to remove His-tag and TEV protease. MALDI-TOF MS was used to analyze the presence of the desired peptide. The desired peptides were concentrated and stored at −20 °C for further application(s).

## MALDI-TOF MS and MS/MS analysis

Mass spectrometry analysis was carried out using either a Bruker UltrafleXtreme MALDI-TOF/TOF mass spectrometer or a Bruker Autoflex speed LRF MALDI-TOF instrument in positive mode. All samples were prepared either by HPLC or desalted using ZipTip C18 or C4 (Millipore) pipet tips prior to analysis. 1 μL of matrix (50 mg/mL solution of 2,5-dihydroxybenzoic acid (DHB) in acetonitrile) was deposited on the target position followed by addition of 1 μL of sample solution onto the same spot and allowed to dry. Mass spectrum data was analyzed using Bruker flexAnalysis 3.4.

## Protein expression and purification

The plasmid used for protein expression was transformed into *E. coli* BL21(DE3) and selected on selected on LB agar plates containing 100 μg/mL ampicillin. A single colony was selected and inoculated into LB containing 100 μg/mL ampicillin at 37 °C overnight. These overnight cultures were then used to inoculate 1.5 L of LB medium with 100 μg/mL ampicillin which were incubated at 37 °C with shaking (150 rpm) until the $OD_{600}$ reached ~ 0.6. Cultures were chilled on ice for 30 min before adding IPTG to a final concentration of 0.3 mM, then, the cultures were placed at 18 °C and shaken at 150 rpm for 16 ~ 20 h. Cells were harvested by centrifugation at 3500 × g for 15 min. The supernatant was decanted, and the cell pellets were used for the following protein purification.

Cell pellets from 1.5 L cultures were suspended in 40 mL of lysis buffer (50 mM Tris-HCl, 500 mM NaCl, 10% glycerol, pH 8.0) and lysed by sonication. The cell lysate was centrifuged at 14,000× g for 30 min at 4 °C, and then the supernatant was loaded onto a 5 mL HisTrap™ HP column, then the column was washed with 50 mL wash buffer (50 mM Tris-HCl, 500 mM NaCl, 25 mM imidazole, pH 8.0) prior to a gradient elution (0–80% elution buffer) of wash buffer and elution buffer (50 mM Tris-HCl, 500 mM NaCl, 500 mM imidazole, pH 8.0) over 40 mL. The fractions were analyzed by SDS-PAGE and target protein-containing fractions were combined and concentrated with appropriate centrifugal filters. CcaF was next desalted by PD-10 with stock buffer (50 mM Tris-HCl, 150 mM NaCl, 10% glycerol, pH 8.0) and stocked at −80 °C for further in vitro leader peptide removal. CcaM, CcaM$_N$ and CcaM$_C$ were further purified by size exclusion chromatography (SEC) using SEC buffer (20 mM HEPES, 300 mM KCl, pH 7.5), the target fraction was collected, concentrated, and stored at −80 °C for further in vitro assay or STD-NMR experiment.

## In vitro enzymatic assay of CcaF, CcaM, CcaM$_N$ and CcaM$_C$

The leader peptide was removed by protease CcaF. The reactions were performed in a 1.5 mL eppendorf tube with optimized ratio (100:1 to 20:1) of different substrates and CcaF in 50 mM Tris-HCl (pH =8.0) buffer at RT overnight. The substrates are different modified full length precursor peptide and synthetic 10-mer peptide. The reaction was monitored by MALDI-TOF MS, if the digestion was not completed, another 50:1 ratio CcaF protease was added and stay at RT for another 8 hours.

In vitro CcaM assay was performed in a 1.5 mL eppendorf tube with optimized ratio (100:1 to 20:1) of different substrates and CcaM with 5 mM ATP, 5 mM MgCl$_2$ and 1 mM DTT in 50 mM Tris-HCl (pH =8.0) buffer at RT. The substrates including full length unmodified CcaA, full-length octa-azole containing peptide CcaA$^{BCH}$, octa-azole containing core peptide cCcaA$^{BCH}$ and different truncated azole containing core peptide cCcaA(1-10)$^{BCH}$, cCcaA(1-20)$^{BCH}$, cCcaA(1-23)$^{BCH}$ and cCcaA(1-26)$^{BCH}$, the reactions were monitored by MALDI-TOF MS.

In vitro CcaM$_N$ alone or CcaM$_N$/CcaM$_C$ pair assay was performed in a 1.5 mL eppendorf tube with 20:1 ratio of octa-azole containing core peptide cCcaA$^{BCH}$ and enzyme in the presence of 5 mM ATP, 5 mM MgCl$_2$ and 1 mM DTT in 50 mM Tris-HCl (pH =8.0) buffer at RT, the reactions were monitored by MALDI-TOF MS.

## Purification of cCcaA$^{BCH}$, cCcaA$^{BCDH}$, cCcaA$^{BCH}$ truncants and carnazolamide

The azole-containing modified peptide was expressed by co-expression of the CccA precursor peptide with the CcaB/CcaC pair. Although the CcaB YcaO alone can catalyze the azoline formation in vitro, the efficiency of the reaction is poor in the absence of the CcaC dehydrogenase. The leaderless modified product was further purified by HPLC after in vitro CcaF assay. The peptides were purified with a Hypersil C18 column (5 μm × 4.6 mm × 250 mm) on Shimadzu LC-20 HPLC instrument. The following mobile phases were used: 0.1% TFA in water (mobile phase A) and 0.1% TFA in acetonitrile (mobile phase B). The samples were centrifuged at 13,000 × g for 10 min to remove any insoluble material before injecting into HPLC.

For cCcaA$^{BCH}$, cCcaA$^{BCDH}$ and cCcaA$^{BCH}$ truncants, method is 0–3 min (25% phase B), 3–20 min (25% to 60% phase B), 20–30 min (25% phase B), flow rate 1 mL/min, UV at 254 nm. For in vitro CcaM dehydration assay, the dehydrated azole-containing core peptides were purified using 0–3 min (30% phase B), 3–20 min (30% to 70% phase B), 20–30 min (30% phase B), flow rate 1 mL/min, UV at 254 nm. The target peptides were collected, verified by MALDI-TOF MS or MALDI-TOF MS/MS, then lyophilized before use in downstream assays or for NMR data collection.

For carnazolamide, the modified precursor peptide CcaA$^{BCDMH}$ was digested by CcaF at RT for 16 hours, the reaction was quenched by heating at 90 °C for 5 mins and centrifuged at 13,000× g for 10 mins, the pellets were extracted by 400 μL methanol for three times, the organic phase and the supernatant were separately for HPLC purification with a Hypersil C18 column (5 μm x 4.6 mm×250 mm) on Shimadzu LC-20 HPLC instrument. The method is as follows: 0–3 min (45% phase B), 3–20 min (45–80% phase B), 20–30 min (45% phase B), flow rate 1 mL/min, UV at 254 nm. Carnazolamide was collected, verified by MALDI-TOF MS and lyophilized before used in downstream Marfey's assays or for NMR data collection.

## NMR spectroscopy

Due to poor solubility in water, carnazolamide (-2.5 mg) was dissolved in 550 μL of CD$_3$OH and transferred into a Wilmad 535-pp NMR tube. NMR data were collected at 25 °C on an Agilent VNMRS 750 MHz NMR spectrometer equipped with a 5 mm indirect-detection triple resonance (HCN) z-gradient probe. One-dimensional (1D) $^1$H NMR, two-dimensional (2D) homonuclear $^1$H-$^1$H TOCSY (total correlation spectroscopy) and $^1$H -$^1$H NOESY (Nuclear Overhauser Effect spectroscopy) spectra were acquired using the Biopack pulse sequences in the VNMRJ 4.2 A software. The spectra were processed with NMRPipe and subsequently analyzed in Sparky and VnmrJ[53,54]. $^1$H chemical shift assignments are shown in Supplementary Table 5.

For the intermediate cCcaA$^{BCH}$, -3 mg cCcaA$^{BCH}$ was dissolved in 550 μL of (9:1 (v/v), H$_2$O/D$_2$O) and transferred into a Wilmad 535-pp NMR tube. NMR data were collected at 25 °C on an Agilent VNMRS 750 MHz NMR spectrometer equipped with a 5 mm indirect-detection triple resonance (HCN) z-gradient probe. One-dimensional (1D) $^1$H NMR, two-dimensional (2D) homonuclear $^1$H-$^1$H TOCSY, $^1$H -$^1$H NOESY spectra and heteronuclear multiple-bond correlation spectroscopy ($^1$H-$^{13}$C HSQC and HMBC) experiments were acquired using the Biopack or Chempack pulse sequences in the VNMRJ 4.2 A software. The spectra were processed with NMRPipe and subsequently analyzed in Sparky and VnmrJ. $^1$H chemical shift assignments are shown in Supplementary Table 4.

For STD (Saturation-Transfer Difference) NMR experiment, -3 mg cCcaA$^{BCH}$ was dissolved in 550 μL of (9:1 (v/v), H$_2$O/D$_2$O) and transferred into a Wilmad 535-pp NMR tube, one dimensional $^1$H NMR was collected as described above. CcaM was purified as described above and buffer exchange into water (pH -5.5) with amicon 30 kDa cutoff concentration tube, 30 μL CcaM stock solution (45 mg/mL) was added into the NMR tube to a final concentration of 20 μM. The STD-NMR

data was collected by irradiating at −1.0 ppm with a 3.0 s saturation time at 25 °C.

## Marfey's assays of cCcaA$^{BCH}$, cCcaA$^{BCDH}$ and carnazolamide

Purified compounds ( ~ 1 mg cCcaA$^{BCH}$, ~1 mg cCcaA$^{BCDH}$ and ~0.25 mg carnazolamide) were separately dissolved in 1 mL 6 M HCl and the samples were heated to 90 °C for 16 hours. After cooling back to room temperature, the solvents were removed using rotary evaporator under reduced pressure, the hydrolysate was re-dissolved in 1 mL water and lyophilized in a 1.5 mL eppendorf tube. The lyophilized hydrolysate was dissolved in 100 μL water and 200 μL acetone containing 1% (W/V) 1-fluoro-2-4-dinitrophenyl-5-L-alanine amide (L-FDAA, Thermo Scientific) and 40 μL freshly prepared 1 M NaHCO$_3$ were added, the reaction vessel was heated at 42 °C for 2 h without shaking using a water bath. The reaction mixture was then quenched using 20 μL 2 M HCl dropwise. Samples were dried under nitrogen gas. L-Amino acid standards were prepared using similar protocol as described above. D-amino standards were generated by reacting the L-amino acid solution with 1-fluoro-2-4-dinitrophenyl-5-D-alanine amide (D-FDAA, Toronto Research Chemicals) with the reported protocol[55].

Derivatized amino acid standards and derivatized hydrolysate were dissolved in 150 μL (85:15 (v/v), H$_2$O/MeCN) 15% MeCN + 0.1% formic acid (LC-MS grade). Samples were vortexed for 5 minutes in and insoluble material was removed by centrifugation at 13,000 × g for 10 min. Amino acid stereochemistry was determined using a Shimadzu LC-MS 2020 with a quadrupole detector and a Macherey-Nagel Nucleodur C18 ec column (250 mm × 4.6 mm, 5 μm particle size, 100 Å pore size). Samples were analyzed using the following conditions: mobile phase A (0.1% formic acid in water, mobile phase B (0.1% formic acid in MeCN), 1 mL/min flow rate, and the following method: 15% B for 7 min, 15-50% B over 45 min, hold at 50% B for 5 min followed by wash and re-equilibration in starting conditions. Analyte (20 μL) was injected and the $[M + H]^+$ ions of the amino acid-Marfey's reagent adducts were selective monitored as follows ($m/z$): Ser, 358; Thr, 372; Asp (amide of Asn hydrolyzed to acid), 386; Ala, 342; mCys (methylated Cys), 388; Glu (amide of Glu hydrolyzed to acid), 400; Val, 370; Pro, 368.

## Bioinformatics

EFI-EST was used to perform analysis of PF13575 (DUF4135, full size = 3,277 members) with the following parameters: alignment score = 80, minimum length 500 aa and E value = −5 to generate a sequence similarity network (SSN)[26]. CcaM and the reported LahM were well separated and classified into different clusters. We then generated a genome neighborhood network (GNN; neighborhood size: 10, minimal co-occurrence percentage lower limit: 5 using the previously generated SSN. Six PF13575 clusters were co-occurred with YcaO. The most abundant cluster 5 has 38 BGCs, all the BGCs were further analyzed by RODEO[56] and NCBI BLAST, *cca* BGC was further chosen for it has 6 more highly similar BGCs which the potential precursor peptides were almost the same.

## Reporting summary

Further information on research design is available in the Nature Portfolio Reporting Summary linked to this article.

## Data availability

All data supporting the findings of this study are presented in the main manuscript text and the Supplementary Information. NCBI, PDB and UniProt accessions are referenced and publicly accessible on the respective websites. Source data are provided with this paper. Data is available from the corresponding authors upon request.

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

## Acknowledgements

This work was supported by the National Institutes of Health (GM079038 to S.K.N.). The Bruker UltrafleXtreme MALDI-TOF/TOF mass spectrometer was purchased in part with a grant from the National Institutes of Health (S10 RR027109 A). We thank Shravan R. Dommaraju for performing the LC-MS experiments. We thank the Tumor Engineering and

Phenotyping Shared Resource (TEP) for helping with the cytotoxicity assay.

## Author contributions

Z.P. and S.K.N. designed the studies. Z.P. performed the experiments, L.Z. acquired and interpreted the NMR data. All authors analyzed data and assisted in the writing and editorial process. S.K.N. conceived of and supervised the project.

## Competing interests

The authors declare no competing interest.
