## [Peer Review File · Nature Communications]

Core-Dependent Post-translational Modifications Guide the Biosynthesis of a New Class of Hypermodified PeptidesEditorial Note: This manuscript has been previously reviewed at another journal that is not operating a transparent peer review scheme. This document only contains reviewer comments and rebuttal letters for versions considered at Nature Communications.

REVIEWERS' COMMENTS

Reviewer #1 (Remarks to the Author):

The authors have improved the manuscript by addressing reviewer comments and highlighting more clearly the novelty of their findings. Upon reflection, owing to the quality of the data and interesting leader-independent enzymology (even if no new modification types were discovered), I am now generally supportive of its publication in Nature Communications with consideration of some minor additional revisions.

Comments:

Line 236 'These data suggested that S-methylation is independent of other modifications.' This statement needs further qualification at this location. 'These data' is ambiguous. The fact that CcaA can be modified by CcaH with or without other modifications present is the rationale for this statement and should be more clearly stated here.

The clarity of some figures could be improved:

Figure 2. For ease of understanding this figure without having to cross-reference other figures, it could be helpful to label product peaks with the number and type of modifications, e.g. 'unmodified', '8x azole + 1x S-Me', etc.

Regarding Reviewer 3's comment '6. Given that the discovery of CcaM is the major finding in this study, the domain composition (shown in the current SI Fig. 2-3) can be shown in a figure in the main text.' The authors responded 'this has been added' but I do not see any domain architectures in the current main text figures.

Figure S4b. In the main text, you mention LahM1 and LahM2 (ref 24) referencing Figure S4, but then these sequences do not appear in the tree and, if present, are not clearly labelled in the alignment. The phylogenetic tree offers some insights into the relationship of different cyclase domains but lacks sufficient detail as there is no associated methods section. How were sequences selected? How was the multiple sequence alignment and tree calculated? Which residues of CcaM were in the trimmed multiple sequence alignment that was used to generate the tree. What are the bootstrap values and cutoffs used; is FlvM1 your outgroup?

Figure 5. It would be helpful to orient the reader to the N- and C-termini in the protein structure images. Please give PDB numbers for CylM and NisC or the reference in the figure caption. Perhaps box the portion of the CcaM prediction in panel a and CylM structure from panel b that are zoomed-in in panel d.

Reviewer #2 (Remarks to the Author):

In this manuscript Nair and coworkers describe the discovery and characterization of a new biosynthetic route to RiPPs that contain dehydrated residues and azoles. This pathway includes an enzyme with a dehydratase domain homologous to those from LanM enzymes

and a cyclase domain homologous to those from LanKC enzymes, which are involved in the biosynthesis of different classes of lanthipeptides. They establish the biosynthetic order of installation of post-translational modifications and show that the LanMbC enzyme is leader independent and dehydrates serine residues preceding azoles. They thoroughly characterize the product of this pathway with MS and NMR.

This leader independence has not been observed for other LanM-like enzymes, nor has the selectivity of modifying residues prior to azoles. The authors thoroughly addressed my previous comments and I believe these findings will be of interest to the community. I recommend this manuscript for publication.

A very minor comment: Have the authors considered the dehydration reaction proceeding through an enamine intermediate in their proposed mechanism?

- Mark Walker

Reviewer #3 (Remarks to the Author):

In this revised manuscript the authors have carefully revised the manuscript to respond to most of comments from the reviewers. In particular, they revised sentences to emphasize the difference between their findings and the previous knowledge (e.g. thiomuracin biosynthesis), and also softened some overextended claims.

Although this reviewer still believes that some nomenclature and discussion described in this paper would be still confusing, this reviewer would respect and accept the authors' responses.

REVIEWERS' COMMENTS

Reviewer #1 (Remarks to the Author):

The authors have improved the manuscript by addressing reviewer comments and highlighting more clearly the novelty of their findings. Upon reflection, owing to the quality of the data and interesting leader-independent enzymology (even if no new modification types were discovered), I am now generally supportive of its publication in Nature Communications with consideration of some minor additional revisions.

Comments:

Line 236 ‘These data suggested that S-methylation is independent of other modifications.’ This statement needs further qualification at this location. ‘These data’ is ambiguous. The fact that CcaA can be modified by CcaH with or without other modifications present is the rationale for this statement and should be more clearly stated here.

Thanks for the suggestion. We have changed it in the main text.

The clarity of some figures could be improved:

Figure 2. For ease of understanding this figure without having to cross-reference other figures, it could be helpful to label product peaks with the number and type of modifications, e.g. ‘unmodified’, ‘8x azole + 1x S-Me’, etc.

Fig. 2 has been changed by adding the description of the modifications.

Regarding Reviewer 3’s comment ‘6. Given that the discovery of CcaM is the major finding in this study, the domain composition (shown in the current SI Fig. 2-3) can be shown in a figure in the main text.’ The authors responded ‘this has been added’ but I do not see any domain architectures in the current main text figures.

Apologies for the unclear response and thanks for the suggestion. We have add the domain architecture of CcaM and structural comparison to clarify the zinc-fingers of CcaM in Fig. 1d.

Figure S4b. In the main text, you mention LahM1 and LahM2 (ref 24) referencing Figure S4, but then these sequences do not appear in the tree and, if present, are not clearly labelled in the alignment. The phylogenetic tree offers some insights into the relationship of different cyclase domains but lacks sufficient detail as there is no associated methods section. How were sequences selected? How was the multiple sequence alignment and tree calculated? Which residues of CcaM were in the trimmed multiple sequence alignment that was used to generate the tree. What are the bootstrap values and cutoffs used; is FlvM1 your outgroup?

We mention the LahM1 and LahM2 in the paper, but they do not belong to the same network with CcaM based on the SSN analysis and these enzymes have not (yet) been functionally

characterized. The sequences were picked from the reported characterized class II and class III lanthipeptide synthetase, domain architecture and AlphaFold prediction were used to determine the cyclase sequence of the selected class II and class III lanthipeptide synthetase. The sequence was aligned using the online Clustal Omega tool, the tree was generated by the default parameters. The cyclase domain (residues 645-1015) of CcaM was used to generate the tree. We don't believe FlvM1 is outgroup because it's sequence identity of the cyclase is low.

Figure 5. It would be helpful to orient the reader to the N- and C-termini in the protein structure images. Please give PDB numbers for CylM and NisC or the reference in the figure caption. Perhaps box the portion of the CcaM prediction in panel a and CylM structure from panel b that are zoomed-in in panel d.

Thanks for the suggestion, we labeled the C-termini of the protein in the structure image as the N-termini of CcaM and CylM was in the back from the orientation we show. Also, we added another structural superposition panel in Fig. 1 according to the suggestion.

Reviewer #2 (Remarks to the Author):

In this manuscript Nair and coworkers describe the discovery and characterization of a new biosynthetic route to RiPPs that contain dehydrated residues and azoles. This pathway includes an enzyme with a dehydratase domain homologous to those from LanM enzymes and a cyclase domain homologous to those from LanKC enzymes, which are involved in the biosynthesis of different classes of lanthipeptides. They establish the biosynthetic order of installation of post-translational modifications and show that the LanMbC enzyme is leader independent and dehydrates serine residues preceding azoles. They thoroughly characterize the product of this pathway with MS and NMR.

This leader independence has not been observed for other LanM-like enzymes, nor has the selectivity of modifying residues prior to azoles. The authors thoroughly addressed my previous comments and I believe these findings will be of interest to the community. I recommend this manuscript for publication.

A very minor comment: Have the authors considered the dehydration reaction proceeding through an enamine intermediate in their proposed mechanism?

Thanks for the suggestion. We are continuing to further study this class of enzymes and a detailed consideration of alternate mechanisms will be proposed in the follow-up study.

Reviewer #3 (Remarks to the Author):

In this revised manuscript the authors have carefully revised the manuscript to respond to most of comments from the reviewers. In particular, they revised sentences to emphasize the difference between their findings and the previous knowledge (e.g. thiomuracin biosynthesis), and also softened some overextended claims.

Thanks.

Although this reviewer still believes that some nomenclature and discussion described in this paper would be still confusing, this reviewer would respect and accept the authors' responses.

Thanks.